# The chicken chorioallantoic membrane model for isolation of CRISPR/cas9-based HSV-1 mutant expressing tumor suppressor p53

**Mishar Kelishadi[1], Hosein Shahsavarani[2,3], Alijan Tabarraei[4,5], Mohammad Ali Shokrgozar[3], Ladan Teimoori-Toolabi[6], Kayhan Azadmanesh[1] ***

**1** Department of Molecular Virology, Pasture Institute of Iran, Tehran, Iran, **2** Faculty of Life Science and Biotechnology, Department of Cell and Molecular Biology, Shahid Beheshti University, Tehran, Iran, **3** Laboratory of Regenerative Medicine and Biomedical Innovations, Pasteur Institute of Iran, National Cell Bank, Tehran, Iran, **4** Infectious Diseases Research Center, Golestan University of Medical Sciences, Gorgan, Iran, **5** Faculty of Medicine, Department of Virology, Golestan University of Medical Sciences, Gorgan, Iran, **6** Molecular Medicine Department, Biotechnology Research Center, Pasteur Institute of Iran, Tehran, Iran

* azadmanesh@pasteur.ac.ir

**Data Availability Statement:** All relevant data are within the paper and its Supporting Information files.

## Abstract

Oncolytic viruses (OVs) have emerged as a novel cancer treatment modality, which selectively target and kill cancer cells while sparing normal ones. Among them, engineered Herpes simplex virus type 1 (HSV-1) has been proposed as a potential treatment for cancer and was moved to phase III clinical trials. Previous studies showed that design of OV therapy combined with p53 gene therapy increases the anti-cancer activities of OVs. Here, the UL39 gene of the ICP34.5 deleted HSV-1 was manipulated with the insertion of the EGFP-p53 expression cassette utilizing CRISPR/ Cas9 editing approach to enhance oncoselectivity and oncotoxicity capabilities. The ΔUL39/Δγ34.5/HSV1-p53 mutant was isolated using the chorioallantoic membrane (CAM) of fertilized chicken eggs as a complementing membrane to support the growth of the viruses with gene deficiencies. Comparing phenotypic features of ΔUL39/Δγ34.5/HSV1-p53-infected cells with the parent Δγ34.5/HSV-1 in vitro revealed that HSV-1-P53 had cytolytic ability in various cell lines from different origin with different p53 expression rates. Altogether, data presented here illustrate the feasibility of exploiting CAM model as a promising strategy for isolating recombinant viruses such as CRISPR/ Cas9 mediated HSV-1-P53 mutant with less virus replication in cell lines due to increased cell mortality induced by exogenous p53.

## Introduction

Cancer is thought to be a global health threat, responsible for one death in six worldwide [1]. Patients with advanced cancer who do not respond to conventional treatments such as surgery, chemotherapy, and radiotherapy have a low overall survival rate. This highlights the need for the development of new therapeutic strategies [2].

**Funding:** This study was funded as Ph.D. student project by Pasteur Institute of Iran (Grant Number: TP-9460). The funders had no role in study design, data collection and analysis, decision to publish, or preparation of the manuscript. The authors received no specific funding for this work.

**Competing interests:** The authors have declared that no competing interests exist.

HSV-1-based recombinant viruses emerged as a new framework in the development of oncolytic viruses due to their capacity for foreign genes, efficient replication, broad host cell range, and safety because their genome is not incorporated into the human genome [3–6].

HSV-1 is a ubiquitous eukaryotic pathogen, belonging to the Herpesviridae family and to the Alphaherpesvirinae subfamily, that has a relatively large enveloped virus with a 152-kb linear double-stranded genome and codes about 80 proteins, half of which are not essential for virus replication [6–8]. The UL39 (ICP6, Infected Cell Protein 6) gene encodes the large subunit of HSV-1 ribonucleotide reductase, a protein complex that converts ribonucleotides to deoxyribonucleotides, providing a major pathway in the synthesis of DNA precursors for its replication. However, it is not crucial for viral growth in dividing cells but its function is required for viral replication and DNA synthesis in quiescent or serum-starved cells as well as neuronal cells. This enzyme is overexpressed in dividing cells such as tumor cells; as a consequence, an ICP6-null mutant preferentially replicates in human tumor cells but not in normal cells with limited dividing activity [9–11].

One of the key features of HSV-1, particularly oncolytic HSV-1 that lacks the neurovirulence factor ICP34.5, is its ability to induce cell death through both apoptosis-dependent and independent mechanisms. This occurs in a cell-specific manner, with caspase-8 playing a crucial role in HSV-1-induced apoptosis. ICP34.5 is a protein expressed by RL1 that inhibits the cellular stress response to viral infection [12–16].

Studies of HSV-dependent apoptosis showed that viral factors such as the US3, US5, ICP4, ICP6, ICP22, ICP27 proteins, glycoprotein D, glycoprotein J, and the latency-associated transcript (LAT) are responsible for the anti-apoptotic activity. Additionally, ICP0 and ICP27 are multifunctional proteins that regulate many aspects of cellular and viral functions, including apoptotic responses [10–12,14–18].

ICP0 has been identified as a pro-apoptotic HSV-1 protein in most studies, however, a few studies have reported that ICP0 can act as an E3 ubiquitin ligase to induce efficient degradation of the p53 protein and inhibit the p53-mediated apoptotic responses in infected cells [10,16–19].

Given the importance of apoptosis in the HSV-1 life cycle, it is crucial to carefully time the induction of apoptosis by the viral genes of HSV-1 when designing therapeutic modalities using this virus [16,17,20,21].

Previous research has demonstrated the crucial role of p53 in apoptosis during HSV1-induced oncolysis. P53 is a regulatory protein and a nuclear transcription factor that plays an essential role in regulating cell division and cell death. In unstressed cells, the expression of p53 is maintained at a low level through ubiquitination and proteasome-mediated degradation of this protein [18,22]. A variety of cellular stresses including DNA damage, hypoxia, oncogene activation, and viral infections lead to the stabilization of p53. By increasing the half-life of p53 and activating transcription of p53-responsive genes, DNA repair and apoptosis are enhanced, which inhibits the propagation of cells with serious DNA damage [18,22,23].

A variety of experimental reports showed that p53 mutations are present in approximately 50% of cancers [20,22,24,25]. In addition, different types of cancer cells exhibiting a nonfunctional p53 pathway which results in the apoptosis pathway deficiency. Therefore, due to the wide interference roles of the p53 in cancer progression, scientists have been seeking feasible and effective experimental platforms to bridge scientific gaps and shed light on mechanistic details of p53 insinuation in cancer cells mortality and affecting the chemo-resistant phenotype in these cells [20,22,24,25].

Based on reported studies, exogenous expression of p53 in human cancer cells during replication of oncolytic viruses such as Vesicular Stomatitis Virus (VSV), Newcastle disease virus (NDV), and adenovirus enhance the cell death leading to anti-tumor effects of these viruses [25–28].

Though HSV-1 mutants were suggested to be a promising candidate for sensitizing the radiotherapy/chemotherapy-resistant tumors [12], some cancer cells (e.g. MCF7 cells) are resistant to HSV-1-dependent apoptosis [10]. It was hypothesized that using HSV-1 oncolytic to restore wild-type P53 activity could be a potential approach to trigger p53-mediated pro-apoptosis and enhance oncolytic potency in advanced tumors.

The CRISPR/Cas9 (Clustered regularly interspaced short palindromic repeats/CRISPR-associated 9) system is a complex of a single guide RNA (sgRNA) that target particular genomic loci and a Cas9 as an endonuclease that makes a double-strand DNA break (DSB). This cleaved DNA subsequently can induce deletions and mutations at the target site or incorporate a transgene into these sites by homologous recombination [29].

In an effort to improve the oncoselectivity and oncotoxicity of HSV-1 as a therapeutic modality, we aimed to inactivate the UL39 gene of a mutant HSV-1, which lacks both copies of the γ34.5 gene. This was achieved through the insertion of a fluorescent P53GFP fusion gene expression cassette using the CRISPR-Cas9 system. We attempted to isolate the recombinant virus from Vero and CAM (as a proof of concept study). Then, we evaluate the oncolytic property of ΔUL39/Δγ34.5/HSV1-p53.

## Materials and methods

### Cell lines, viruses, plasmids and chicken eggs

In this study, Vero (African green monkey kidney, NCBI-C101), BHK-21 (Baby hamster kidney, NCBI-C107), A549 (human lung epithelial, NCBI- C137), MDA-MB-468 (Human Adenocarcinoma, NCBI- C208), Hela (Human cervical carcinoma, NCBI- C115), HEK 293 (Human Embryo Kidney, NCBI- C497), HEK 293T (Human Embryonic Kidney, NCBI-C498), Caco-2 (Human Colorectal Adenocarcinoma, NCBI- C139), and NIH3T3 (Mouse Embryo cell, NCBI- C156) cell lines were obtained from the National Cell Bank of Pasteur Institute of Iran. These cells were cultured in high glucose Dulbecco's modified Eagle's medium (DMEM) (Gibco, Germany) supplemented with 10% heat-inactivated Fetal Bovine Serum (FBS; Gibco), 2mM L-glutamine, and 1% Penicillin/Streptomycin (Gibco, Germany) at 37˚C with a humidified atmosphere containing 5% CO2. All cell lines were tested to be contamination-free.

The maternal virus used in this study was Δγ34.5/HSV-1 virus, an ICP34.5-null mutant HSV-1 in which both copies of the γ34.5 gene were replaced by an insert carrying a Blecherry reporter gene (a red fluorescent protein) driven by the cytomegalovirus promoter [30,31].

pIRES2-EGFP plasmid (Addgene, #6029–1), sgRNA/Cas9 cloning vector pX459-puro (Addgene, #62988), sgRNA/Cas9 cloning vector pX459-mCherry (Addgene, #64324), the pIRES2-EGFP-p53 WT Plasmid (Addgene, #49242).

Ten-day-old Specific Pathogen Free (SPF) embryonated chicken eggs were purchased from Razi Vaccine & Serum Research Institute (Karaj, Iran). As it is generally accepted that the embryo cannot feel pain until approximately day 19, special permission for animal experiments was not required [32–37].

### Δγ34.5/HSV-1 preparation and DNA extraction

Vero cells were infected with Δγ34.5/HSV-1 at a multiplicity of infection (MOI) of 0.01. After 2 days, the cells were harvested after observation of the total cytopathic effect. The supernatant was titrated, aliquot, and stored at -80˚C.

DNA was purified from virus-infected Vero cells stock by a commercially available kit (High Pure Extraction Kit; Roche Diagnostics GmbH, Mannheim, Germany) according to the manufacturer's instructions.

## Design and cloning gRNA oligo's into Cas9 vector

The CRISPR/Cas9 system used in this study was constructed by introducing synthesized oligo primers targeting the UL39 gene of Δγ34.5/HSV-1 into sgRNA/Cas9 cloning vector pX459 according to Feng Zhang's lab recommendations (PX330 cloning protocol): (https://www.addgene.org/crispr/zhang/)(https://pharm.ucsf.edu/sites/pharm.ucsf.edu/files/xinchen/media browser/PX330%20cloning%20protocol.docx). Briefly, the gRNAs were designed and selected using available online tools (http://crispr.mit.edu), https://chopchop.cbu.uib.no/ and (http://www.rgenome.net/cas-offinder) to select the optimal sequence for maximizing double-stranded breaks (DSBs) while minimizing the off-target effect.

Two complimentary oligodeoxynucleotides gRNA F and gRNA R (Table 1) were annealed in a thermocycler and the resulting dsDNA fragment was then ligated into the *Bbs*I (Thermo-Fisher Scientifc, USA) site of linearized pX459. The resulting plasmid is named Cas9/gRNA$_{UL39}$. The insertion of the gRNAs was confirmed with the test primers (CMV$_P$F and gRNA R) (Table 1) and sequencing.

## Generation of UL39 shuttle donor vector for homologous recombination (HR)

UL39 shuttle donor vector was constructed based on pJET1.2/blunt Cloning Vector backbone, a high-efficiency TA cloning vector (CloneJET PCR Cloning Kit, ThermoFisher Scientific, USA). The full sequence of the expression cassette containing the CMV promoter, EGFP coding region, and the SV40 early mRNA polyadenylation signal was amplified from pIRES2-EGFP plasmid using the forward primer (CMV$_P$ F) containing *Eco*RI restriction site and the reverse primer (PolyA R) containing *Mlu*I (ThermoFisher Scientifc, USA) restriction site.

The resulting PCR product was purified using GeneJET PCR Purification Kit (Thermo Fisher Scientific, USA) according to the manufacturer's manual and directly cloned in the pJET1.2/blunt cloning vector to generate GFP-pJET plasmid (Table 1).

Right and left homologous fragments encoding about 856 and 823 nucleotides of the genome which flank the specific regions within the UL39 gene of Δγ34.5/HSV-1 genome were amplified by PCR with the primers; Upstream homologous arm UL39 forward) containing *Bgl*ll (ThermoFisher Scientifc, USA) restriction site and upstream homologous arm UL39 reverse containing *Eco*RI (ThermoFisher Scientifc, USA) restriction site for UL39 R fragment and with the primers downstream homologous arm UL39 forward containing *Mlu*I restriction

**Table 1. Primer sequences for generating homologous fragments of UL39 and recombination analysis.**

| Target amplicon | Sequence (5'-3') Forward | Sequence (5'-3') Reverse | Annealing temperature (˚C) | Product size (bp) |
|---|---|---|---|---|
| Upstream homologous arm UL39 | tatcagatctGGTGGTCCCTCAGCG | ccgaattcACTTGAACATTTCCCACCAC | 61 | 856 |
| Downstream homologous arm UL39 | tatcacgcgtgGCCATGCTGAACCTG | ctcctgcagTGTTCACCATCAGCAC | 61 | 823 |
| CMV$_P$ F- EGFP—Poly A R | ccgcgggaattcTAGTTATTAATAGTAA | gatatcacgcgtTAAGATACATTGATGAGTT | 61 | 2190 |
| SgRNA (UL39) | caccgTCTGGTGGTCGTAGAGGCGG | aaacCCGCCTCTACGACCACCAGAc | 59 | 22 |
| UL39 test | ACGACTTTGGGCTTCTCAAC | CCTTGTTTGTGGTGGCCTGG | 61 | 671 |

site and the reverse primer Upstream homologous arm UL39 reverse containing *Pst*I (ThermoFisher Scientifc, USA) restriction site for UL39L fragment. After purification, the fragments were sub-cloned sequentially into the GFP-pJET at *Mlu*I/*Pst*I and *Bgl*ll/*Eco*RI sites. The resulting plasmid, also named pJET-UL39R-CMV-GFP-UL39L was confirmed by Sanger sequencing (Table 1).

The p53 coding sequence in the pIRES2-EGFP-p53WT plasmid was cut out with *Nhe*I (ThermoFisher Scientifc, USA) and *Not*I (ThermoFisher Scientifc, USA) and ligated into the similar restriction sites of pJET-UL39R-CMV-GFP-UL39L shuttle plasmid to generate pJET-UL39R-CMV-GFP-P53-UL39L.

## Plasmids extraction and purification

Plasmid isolation and DNA fragments purification was performed using the ThermoScientific GeneJET Plasmid Miniprep Kit (Thermo Scientific, Waltham, MA USA) and GeneJET Gel Extraction Kit (Thermo Scientific, Waltham, MA USA), regarding the manufacturer's instructions.

## Generation of ΔUL39/Δγ34.5/HSV1-p53 mutant using the CRISPRCas9 system

According to the manufacturer's protocol, transient transfections were carried out on BHK cells using ScreenFect™ A plus (Fujifilm WAKO, Japan) according to the manufacturer's protocol. The reagent consisted of 1μg of DNA and 1μL of reagent per each well of a 24-well plate. SgRNA/Cas9 cloning vector pX459-mCherry (Cat no. 64324; Addgene) and pJET-UL39R-GFP-p53-UL39L plasmid were used as controls in all transfection experiments for monitoring the cell viability along the reporter gene expression.

Briefly, BHK cells were seeded into 24-well plates (SPL Life Sciences, Korea) at a density of $0.05 \times 10^6$ cells/well to achieve 80% confluency for transfection. Cells were then transfected with Cas9/gRNA$_{UL39}$ plasmid, which contains a puromycin-resistance gene, in the presence of 1 μM SCR7, a non-homologous end joining (NHEJ) inhibitor (Sigma-Aldrich, USA). After 24 hours, a second transfection was performed using the pJET-UL39R-GFP-p53-UL39L plasmid, supplemented with 10 ug/mL of puromycin (Bio Basic, Canada). 48 hours later, the cells were inoculated with Δγ34.5/HSV-1 at an MOI of 1 and incubated for 1–2 hours at 37˚C with 5% $CO_2$. The inoculum was removed and the cells were gently washed with the pre-warmed PBS to remove any un-adsorbed input virus. The transfection/infection supernatant from BHK cells was harvested 24–48 hours later, upon the appearance of an 80% cytopathic effect on the cell monolayer. After three freeze-thaw cycles, aliquots were stored at -80˚C (Fig 1).

## Isolation of the ΔUL39/Δγ34.5/HSV1-p53 mutant in vitro

96-well plated were seeded with $10^4$ Vero cells per well, one day before infection. To isolate the recombinant viruses, confluent monolayers of cells were infected with different dilutions of the mutant viral supernatant. The plates were daily monitored using an inverted fluorescent microscope (Nikon eclipse Ti-S, Japan) for EGFP expression at 1–3 days post-infection.

## Isolation of the ΔUL39/Δγ34.5/HSV1-p53 mutant in CAM of the fertilized chicken eggs

To isolate the recombinant virus in the CAM, eggs were first checked for viability and a false air sac and window opening were created in the shell. The CAMs were then inoculated with 0.1 ml of serially 10-fold diluted viruses (the transfection/infection supernatants) each [36,38–40].

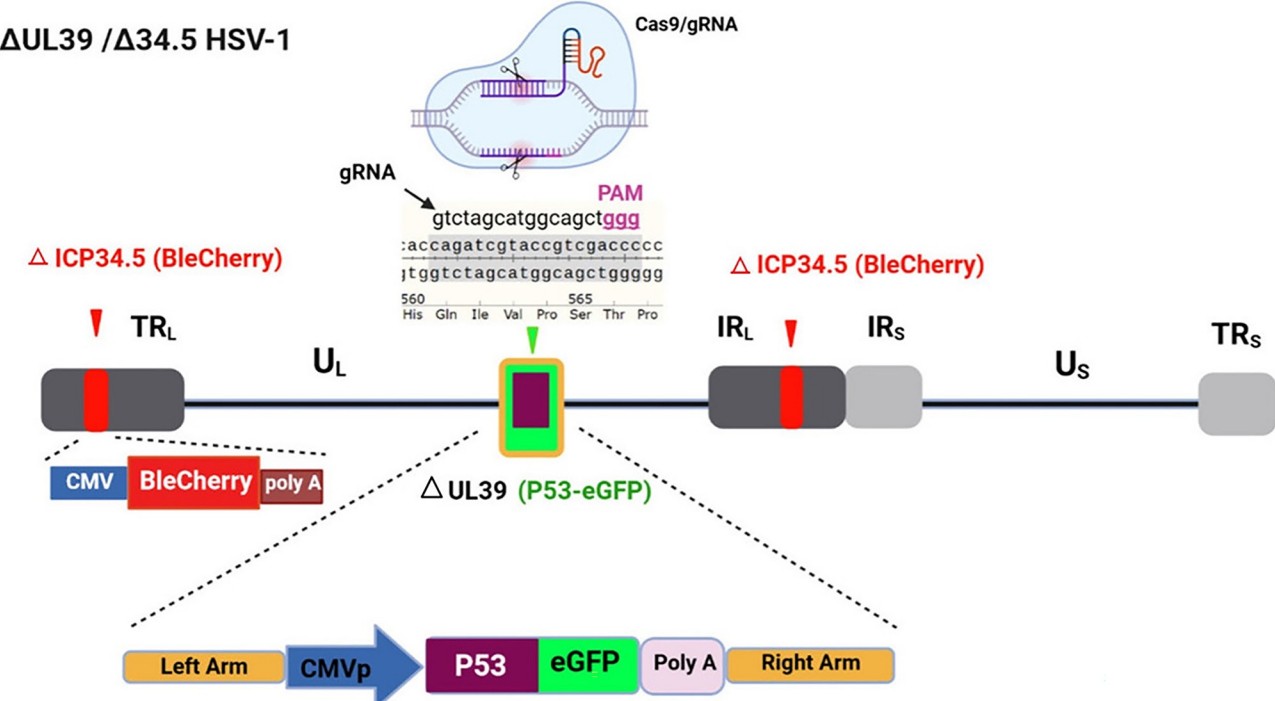

**Fig 1. Schematic structure of ΔUL39(green)/Δ 34.5(red) HSV-1-p53 that was developed in an ICP34.5 deleted HSV-1 backbone.** Modifications include deletion of the UL39 gene and insertion of an EGFP-P53 expression cassette utilizing a CRISPR/Cas9-mediated editing genome system. The internal ribosome entry site (IRES2) sequence between p53 and GFP genes allows the co-expression of the genes and facilitates their detection in mammalian cells while retaining the properties of wild-type p53.

The opening was sealed with povidone iodine-impregnated paraffin wax to prevent contamination. The eggs were then incubated in a Small Egg incubator Easy Bator 3 (Eskandari Industrial Group, Iran) at 34.5°C and 58–60% humidity, without rotation. After 72 hours, the eggs were placed in a refrigerator at 4°C for 2 hours to constrict the blood vessels before harvesting the CAMs [41,42]. The harvested membranes were placed in a petri dish containing normal saline supplemented with 4% penicillin/streptomycin to flatten the rolled CAM. Then DMEM media was then gently removed and the pocks were analyzed using an inverted fluorescent microscope for eGFP or BleCherry expression.

For purification of the HSV1-P53 mutant, the pocks which were positive for both BleCherry and GFP signals (pocks containing the recombinant green/red virus) were isolated. Thereafter, they were dispersed by trypsin and vortexed and after three freeze-thaw cycles, they were subcultured in the new CAM (Fig 2).

## PCR analysis and sequencing for verification of homologous recombination

To verify recombination, the pocks were positive for both BleCherry and GFP signals (pocks containing the recombinant green/red virus) were isolated for DNA extraction using High Pure Viral Nucleic Acid Kit) Roche Diagnostics GmbH, Mannheim, Germany) according to the manufacturer's instructions. PCR was performed using a PCR master mix kit (Taq DNA Polymerase Master Mix RED 2x, Ampliqon, Denmark) in a total volume of 25 containing of 12.5 μl Taq DNA Polymerase, 1x Master Mix RED, (~100–150 ng) of HSV-1 DNA and 0.2 μM of each forward and reverse test primers. The sequences of primers are given in Table 1. The

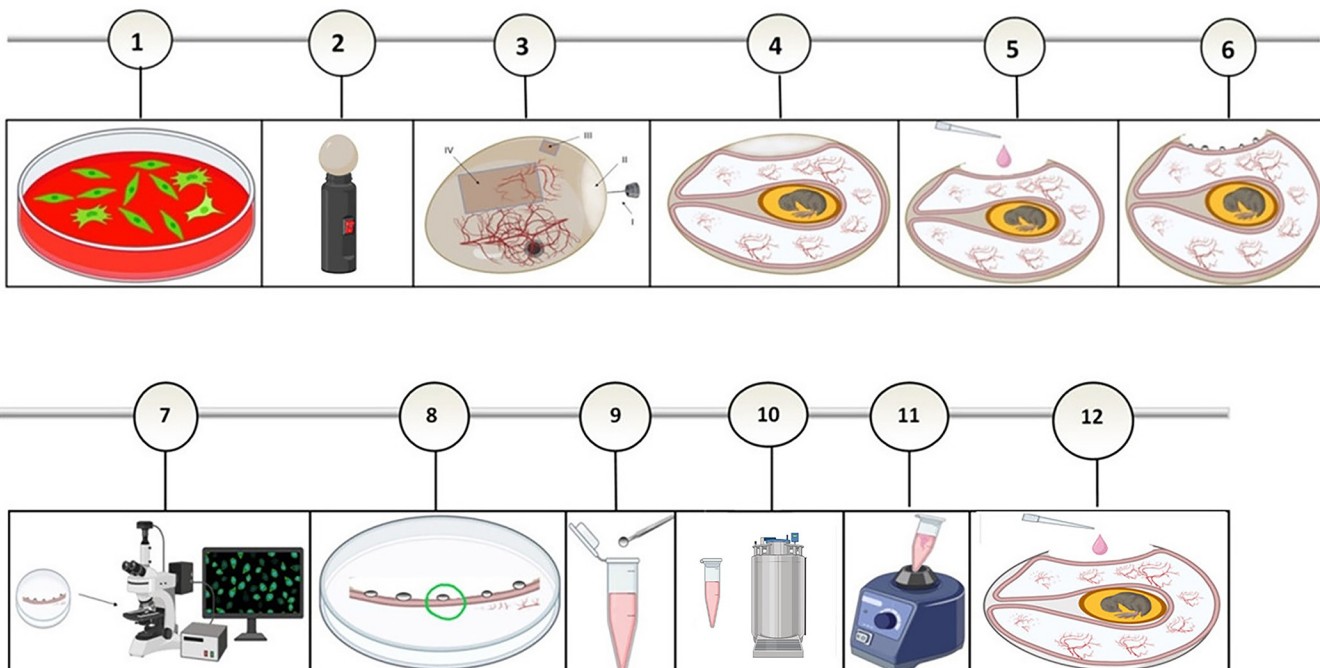

**Fig 2. Experiment workflow for generation of recombinant virus using chorioallantoic membrane (CAM) of fertilized chicken eggs.** The process of transfection/infection of cells (1.2). Candling egg for visualizing the air sac (II) and identification of the egg vasculature (2.2). Preparation of artificial air sacs including the major steps; markings (I, III, IV), drilling (I) and creation of a square opening (III) and an operating window (IV) in the egg shell (3.2). Generation of the artificial air sac (4.2). Inoculation of the diluted virus onto the CAM membrane (5.2). The established pocks on the CAM membrane three days after the virus inoculation (6.2). Analyzing of the pocks using an invert fluorescent microscope for fluorescence protein expression (7.2). Selecting of the pock with fluorescence signals (8.2). Dipping the selected pocks into the culture medium (DMEM media) (9.2). Three freeze–thaw cycles (10.2). Vigorous vortexing of the media containing the pock (11.2). Inoculation of the pock (virus) onto a new CAM (12.2).

program of PCR was as follows: 95˚C for 5 min; 30 cycles of 95˚C for 60 s, 55˚C for 30 s, and 72˚C for 2 min followed by the final extension step at 72˚C for 5 min. The PCR products were loaded on a 1% agarose gel and visualized by exposing it to ultraviolet (UV) light.

### Phenotypic characterization of ΔUL39/Δγ34.5/HSV-p53mutant

Vero, BHK-21, A549, MDA-MB-468, Hela, HEK 293, HEK 293T, Caco-2, and NIH3T3 cell lines ($10^4$ cell/well) were seeded into 96-well plates to characterize the newly isolated oncolytic virus phenotypically. They were infected the following day with 0.1 ml of various dilutions of the recombinant virus. After 1 hour of incubation at 37˚C, cells were cultured in DMEM medium (containing 1% FBS) and were incubated for up to 5 days for GFP expression. All assays were performed in triplicates.

## Result

### Generation of ΔUL39/Δγ34.5/HSV-p53 and verification of homologous recombination

For the generation of the recombinant virus, the sgRNA targeting the specific regions within the UL39 gene of the Δγ34.5/HSV1 was cloned into sgRNA/Cas9 cloning vector pX459.

The donor vector containing the UL39R-GFP-p53-UL39L fragment was used for homologous recombination to improve the efficiency of homology-directed repair (HDR). Both Cas9/gRNA$_{UL39}$ and the donor plasmids were transfected into BHK-21 cells.

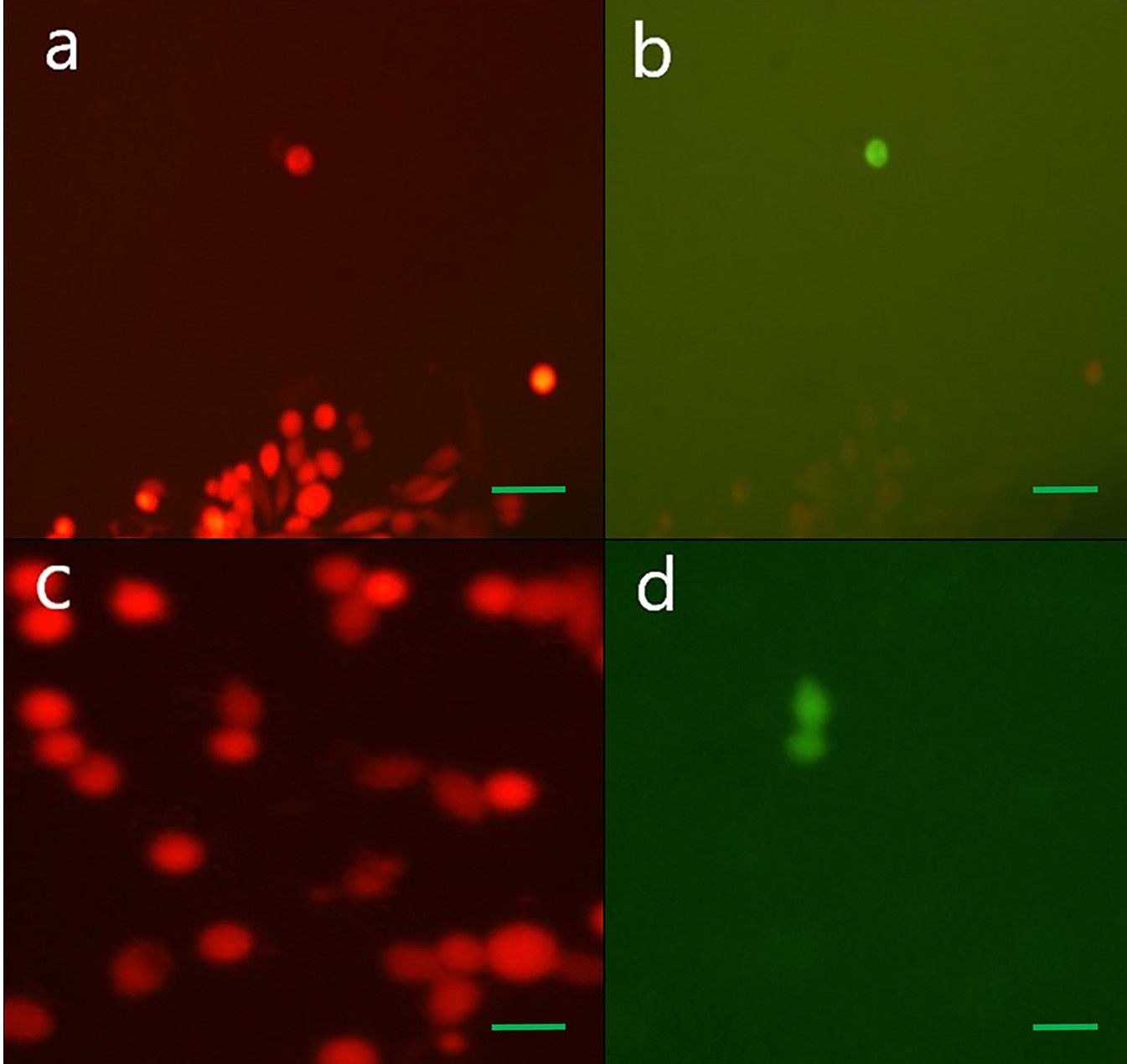

**Fig 3.** Representative results of inoculation of the viral supernatant that were expected to contain mutated viruses in BHK-21 (a, b) and MDA-MB-468 (c, d) cell lines. The recombinant virus replication was limited to a single cell. (a-d) 200X; scale bar: 100 μm.

The viral supernatant containing HSV-1 mutant was inoculated in Vero and BHK-21, MDA-MB-468, Hela and HEK 293T cell lines to isolate the recombinant virus but in all of them, the recombinant virus replication was limited to a single cell (Fig 3). So the viral supernatant was inoculated onto CAM (Fig 4).

Of note, high titer of the virus in the inoculum ($10^7$ pfu/ml) induced no discrete pocks in CAM and the CAM membrane looked normal but fluorescent analysis of red Δ34.5/HSV-1 illustrated red confluent lesions in the CAM, reflecting the virus distribution throughout the CAM (Fig 5).

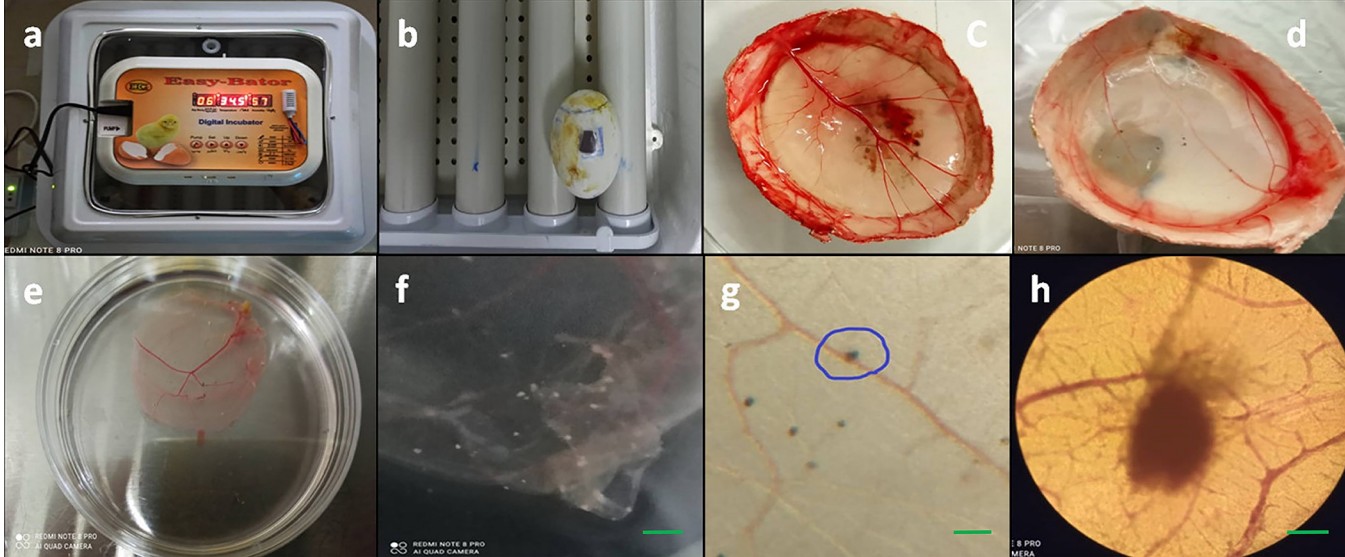

**Fig 4. Representative results of using chorioallantoic membrane (CAM) of fertilized chicken eggs for generation of the recombinant virus.** Incubation of the eggs in small egg incubator Easy Bator (a). The generated operating window in the egg shell (b). The harvested CAMs three days after inoculation of the virus with a titer of $1 \times 10^4$ plaque-forming units (pfu)/ml (c) and $1 \times 10^2$ pfu/ml (d), illustrating visible pocks. The isolated membrane from the egg shell in a petridish (e). Stereomicroscope image of pocks formed on CAM (f) 40X; scale bar: 100 µm. Microscopic image of pocks formed on CAM (g) 100X; scale bar: 100 µm. Higher magnification of the pock (h) 200X; scale bar: 100 µm, Brightfield).

Three days later, the pocks positive for both BleCherry and GFP signals were confirmed by visualizing by the inverted fluorescent microscopy (Fig 6) and molecular analysis (Fig 7).

### In vitro characterization of the edited ΔUL39/Δγ34.5/HSV1-p53 mutant

Non-cancerous cell lines (Vero; wild-type p53, BHK-21; wild-type p53, NIH3T3; wild-type p53) and cancerous cell lines (HEK 293; wild-type p53 and immortalized by Ad 5 E1A and

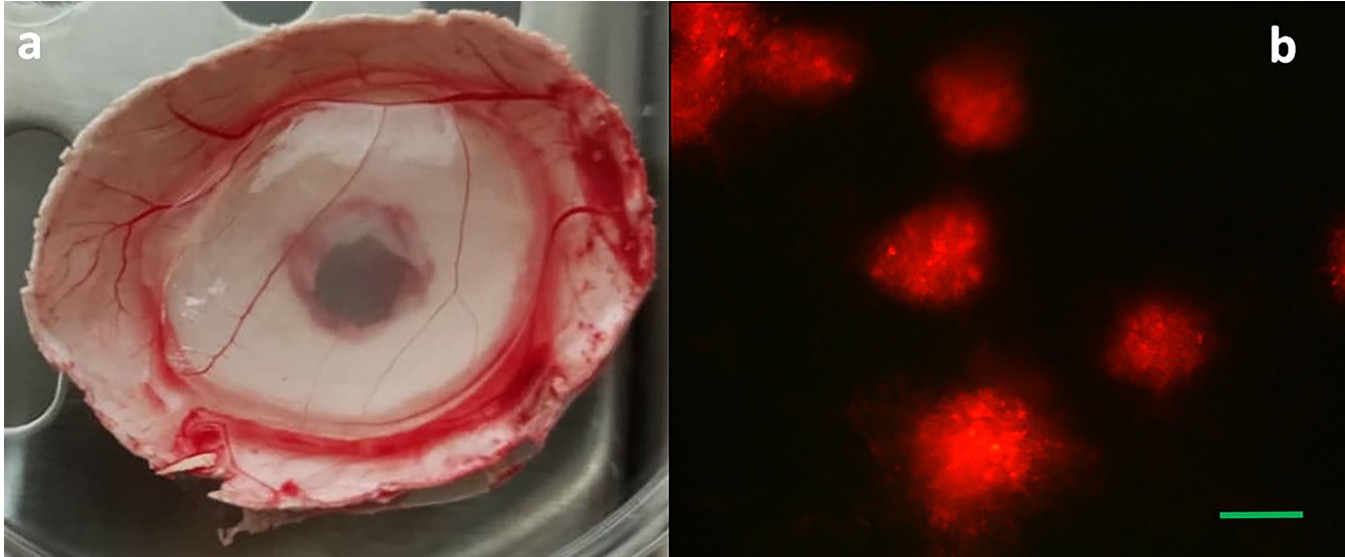

**Fig 5. High titer of the virus resulted in confluent lesions and induced no discrete pocks in CAM.** Macroscopic picture of high titer virus -infected CAM (a). Microscopic picture of high titer virus -infected CAM (b) 200X; scale bar: 100 µm.

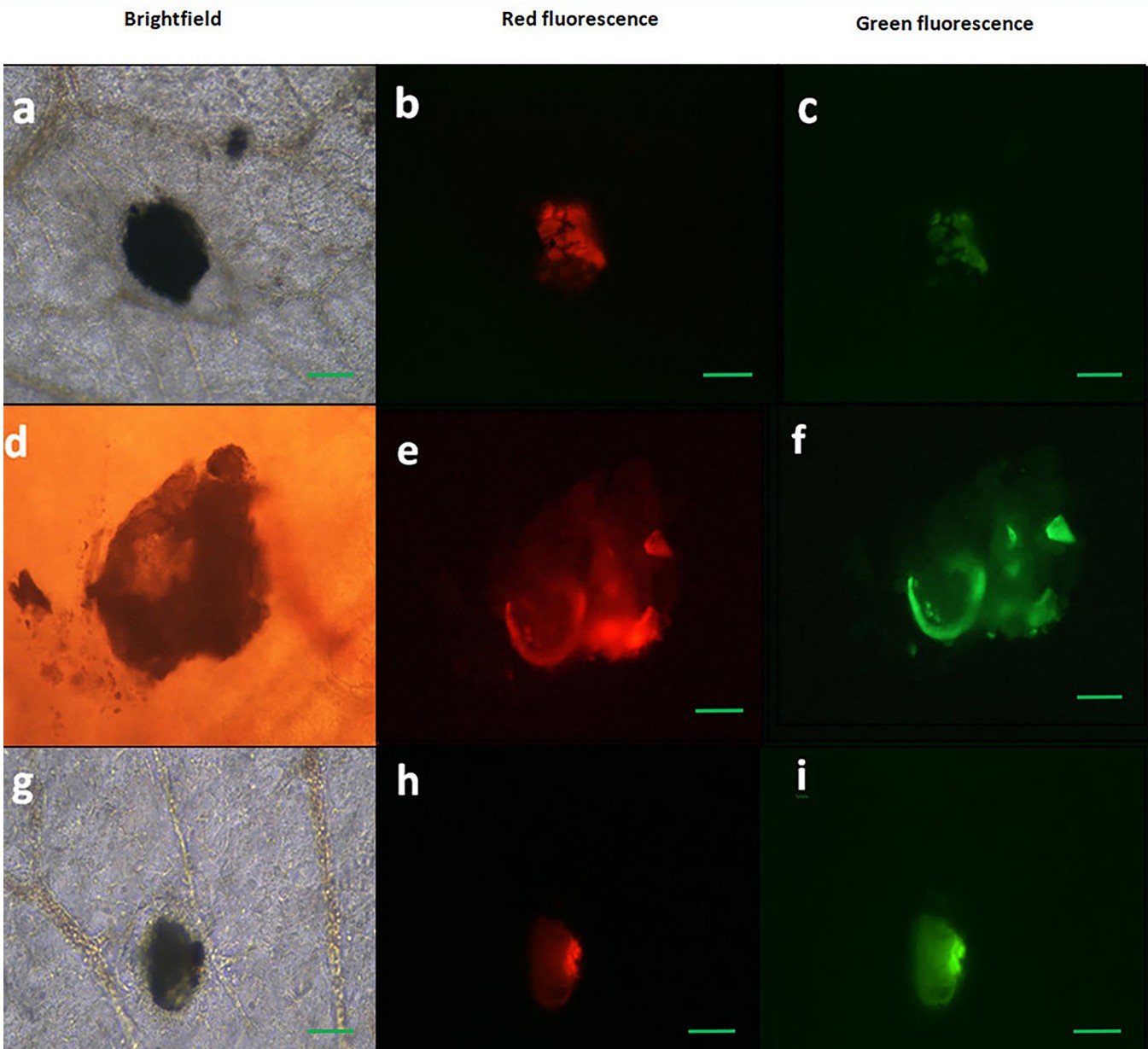

**Fig 6.** Fluorescence and brightfield images of representative pocks showing the EGFP & BleCherry signals, confirming generation of the ΔUL39 (green)/Δ 34.5 (red) HSV1-EGFP-p53 and (a,b, c, d, e and f) and the ΔUL39 (green)/Δ 34.5 (red) HSV1-EGFP, as a control virus, (g, h and i) in chorioallantoic membrane (CAM) of fertilized chicken eggs. (a-i) 200X; scale bar: 100 μm.

E1B, HEK 293T; wild-type p53 that also expresses SV40 large T antigen, A549; wild-type p53, MDA-MB-468; mutant p53 and P53- resistant, HeLa; wild-type p53 in which p53 is strongly repressed by overexpression of E6 protein from oncogenic HPV, Caco-2; mutant p53 (infected with ΔUL39/Δγ34.5/HSV-p53 displayed different phenotypes from the parental Δ34.5/HSV-1 virus (Fig 8) and had cytolytic ability in the tested cell lines that are of various tissue origins and with different p53 status (Vero, BHK-21, A549, MDA-MB-468, Hela, HEK293, HEK293T, Caco-2, and NIH3T3 cell lines). Cultivation of ΔUL39/Δ γ34.5/HSV-p53 mutant in these cell

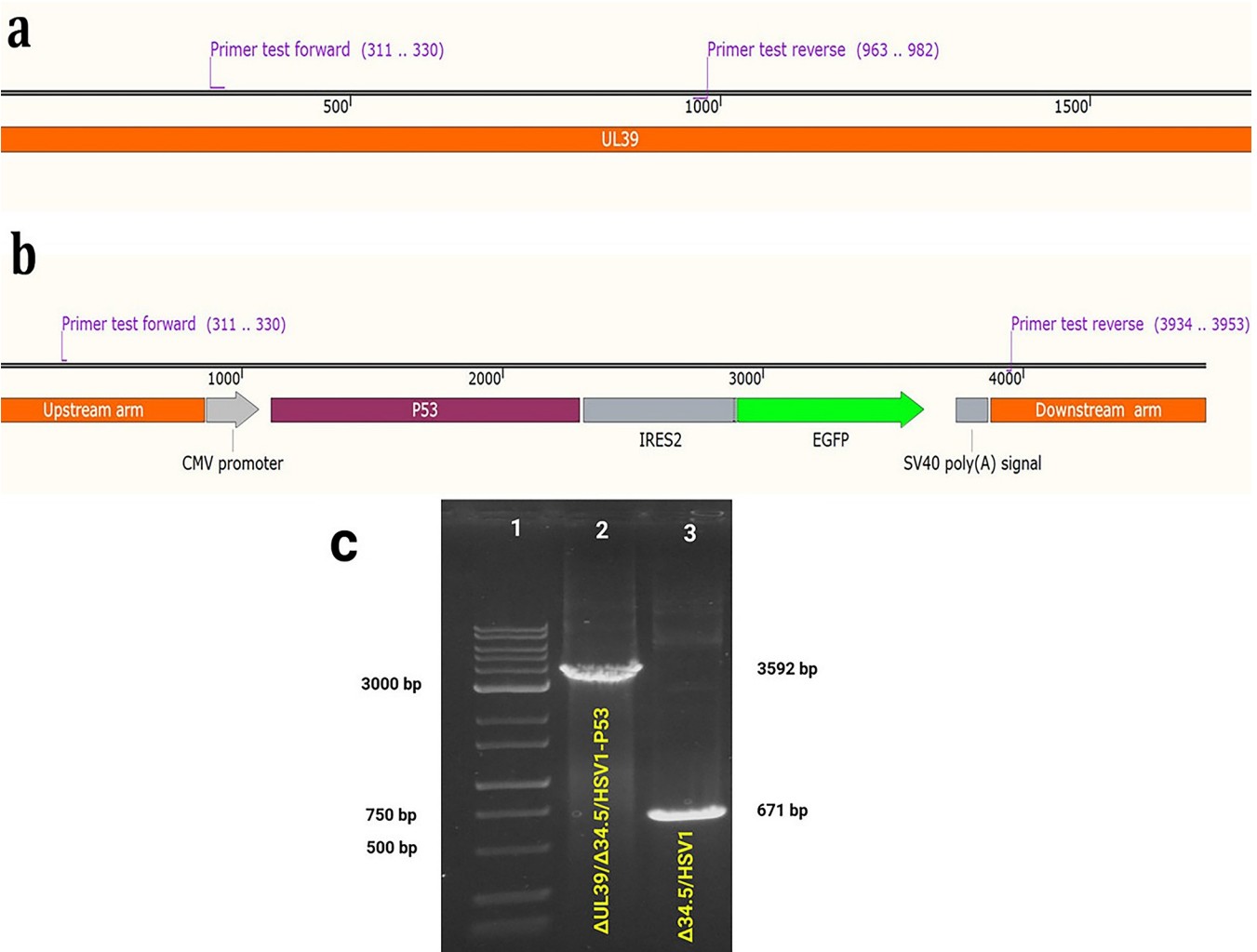

**Fig 7. Schematic illustration of the test primers binding site to UL39 in.** Δ34.5/HSV-1 parental virus (a) recombinant ΔUL39/Δ34.5/HSV1-P53 virus (b). Confirmation of ΔUL39/Δ34.5/HSV1-P53 generation by PCR (c): PCR product of the test primers in Δ34.5/HSV-1 indicates an intact band including upstream and downstream sequences of UL39 gene (lane 3; 671bp), while in ΔUL39/Δ34.5/HSV1-P53, a band (lane 2; 3592 bp) including upstream and downstream sequences of deleted the UL39 coding sequence (CDS) plus CMV-EGFP-P53-polyA is shown (lane 1, CinnaGen 100 bp DNA ladder).

lines was also associated with rounding of cells in early infection without viral cell-to-cell spread and loss of adherence to the monolayer (Fig 9).

## Discussion

Present study aimed to improve the oncoselectivity and oncotoxicity properties of a neuroattenuated Δγ34.5/HSV-1 mutant by manipulating the UL39 gene. This was achieved by inserting an EGFP-p53 expression cassette under the control of the CMV promoter using the CRISPR-Cas9 system, taking advantage of P53's ability to trigger apoptosis in target cancer cells.

In early experiments, the recombinant ΔUL39/Δγ34.5/HSV-p53 was isolated by cultivating the transfected/infected viral supernatant in Vero and BHK cell lines, which are commonly used in HSV research. The viral supernatant was also cultivated in cell lines that have been shown to use strategies to manage p53 signaling in favor of their continued survival, such as

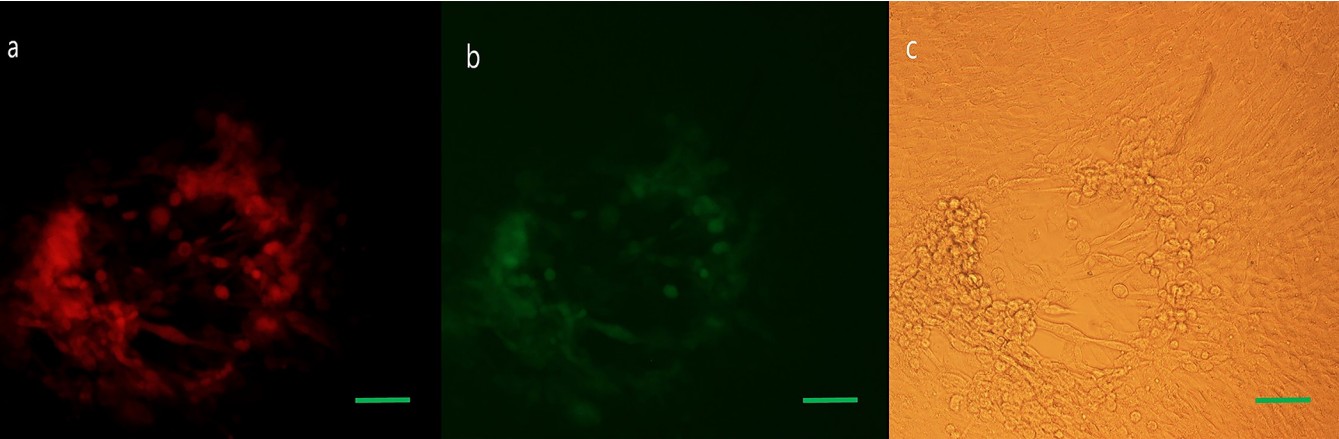

**Fig 8. ΔUL39/Δγ34.5/ HSV1-GFP -infected BHK cells as controls.** Fluorescence and brightfield imaging was done 72 h postinfection. (a-c) 100X; scale bar: 100 μm.

MDA-MB-468, Hela, and HEK 293T cell lines. Cultivation of the viral supernatant in these cell lines was restricted to a single round of replication without viral cell-to-cell spread (Fig 3).

A theoretical explanation may be that the disruption of the UL39 gene (as a viral anti-apoptotic factor) [19] and p53 overexpression during replication of ΔUL39/Δγ34.5/HSV-p53 activate both intrinsic and extrinsic pathways of apoptosis in the virus- infected cancer cells accelerating the premature death, thereby limiting the virus replication to an abortive infection in cells. Hence, we deduced that the defective virus needs to be replicated in a complementary permissive and p53-resistant cellular model [9–11,43].

In 2012, a few publications reported the ability of pock forming by HSV-1 on the CAM of fertile hens' eggs in which pocks were white, superficial, and separate that remained small [35,36,38]. Although the embryonated hen's egg is largely supplanted by the cultured cells for the isolation and cultivation of viruses, this system is newly used for the in vivo analysis of oncolytic viruses and investigating several functional features of tumor biology such as angiogenesis, cell invasion, and metastasis [33,35,36,44].

Supplying the essential enzymes during the highly active metabolism of tissues in the embryo development and proliferation of cells in pocks on the virus-infected CAM could be used for replication of the virus as well [38,39,44,45]. Besides, the probable lesser sensitivity of CAM cells of chick embryos to the human p53 effects [15,46] led us to the hypothesis that the CAM model might be a suitable tissue source for ΔUL39/Δ34.5/HSV1-p53 propagation in which the virus can borrow the Ribonucleotide Reductase (RNR) essential enzyme from the host cells and replicate efficiently.

Despite the interesting observations in ΔUL39/Δγ34.5/HSV-p53 multiplication on the CAM, our work highlights the limitations of replication of CAM-adapted HSV-1-p53 in vitro culture. It was suggested that integrating the p53 gene into the viral genome can make cells more susceptible to premature death during virus replication. This limits virus multiplication to a single round of replication, resulting in the production of replication-defective mutant viruses.

Black et al. showed that VSV-encoded p53 inhibited virus replication in non-malignant human pancreatic ductal cells [47]. As well, the finding of other researchers showed the VSV and Adenovirus carrying the p53 gene can simultaneously assist virus replication while enhancing oncolytic potency in cancer cells such as lung carcinoma cells, breast carcinoma cells, cervix carcinoma cells, prostate carcinoma cells, and pancreatic cancer cells [20,25,47–50].

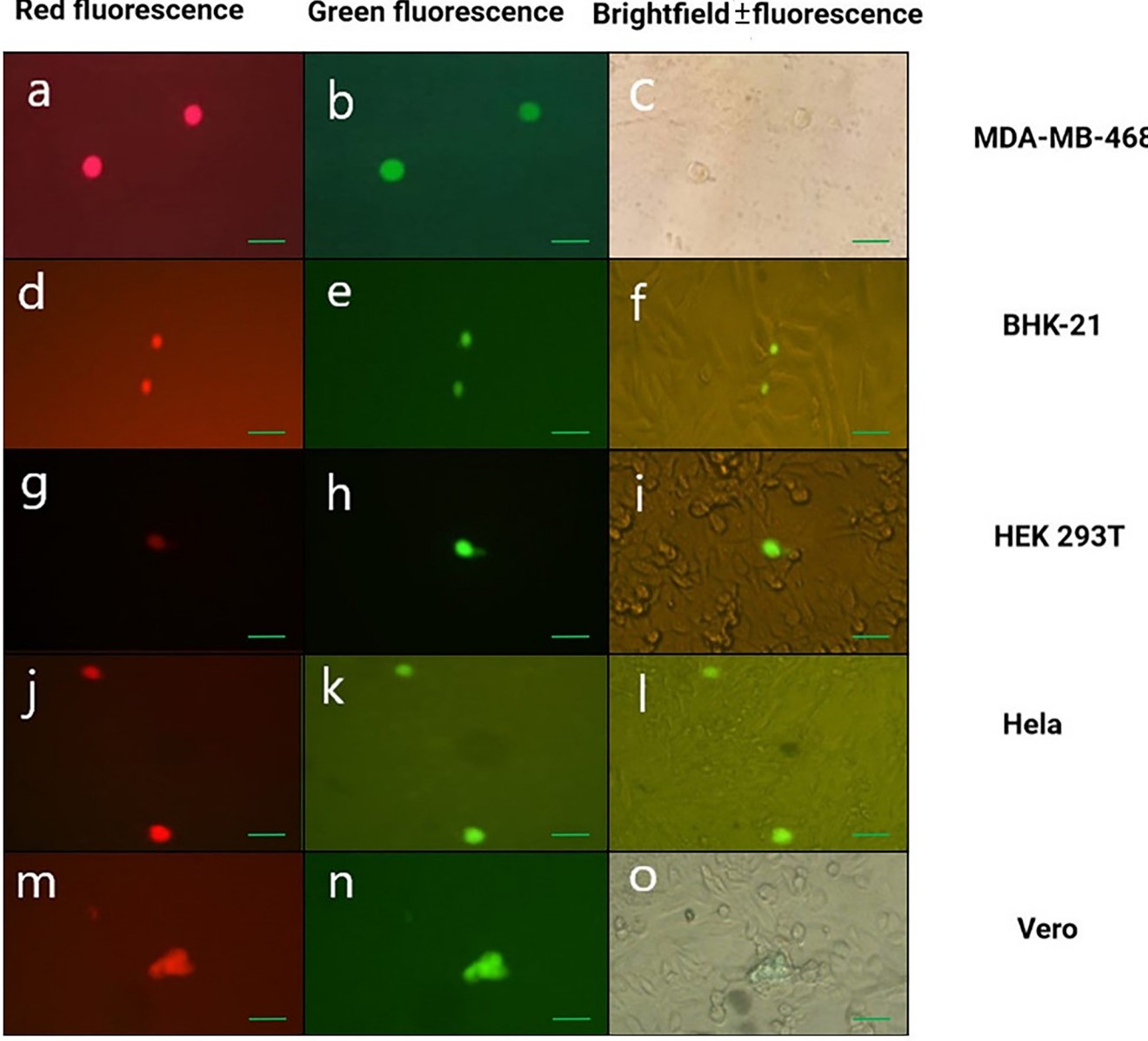

**Fig 9. Representative images of cultivation of CAM-adapted HSV-p53 mutant in the cell lines.** Fluorescence images of cell lines showing the EGFP & BleCherry signals, confirming generation of the ΔUL39 (green)/Δ 34.5 (red) HSV1-p53) a, b, d, e, g, h, j, k, m and n) that associate with rounding of cells in early infection without viral cell-to-cell spread and loss of adherence to the monolayer. Brightfield image of MDA-MB-468 (c). Brightfield & fluorescence images of BHK-21, HEK 293T, Hela, Vero cells (f, i, l and o). (a-o) 200X; scale bar: 100 μm.

Ying et al. reported that the Recombinant Newcastle disease virus expressing P53 (rNDV-P53) has no distinctions in the kinetics and magnitude of replication compared with the vehicle virus (rNDV) in hepatocellular carcinoma model [26].

However, during productive HSV-1 infection, a balance between the pro-apoptotic and anti-apoptotic factors is set up that allows virus propagation. At first, apoptosis is initiated by the immediate early gene expression and later, it is modulated by the early and late viral anti-apoptotic genes which block apoptosis progress [10,14,21,46,51,52].

p53 is generally considered an inducer of apoptosis in many viruses-infected cells, however the previous reports suggested that during HSV-1 infection, p53 plays both positive and negative roles in HSV-1 replication; upregulating ICP27 expression, early during the infection and downregulation of ICP0 at later stages of infection, inhibit apoptosis during the HSV infection. In addition, p53's positive and negative effects in HSV-1-infected cells are organized by multiple mechanisms in a time-dependent and p53 status-dependent manner. However, a threshold of caspase activation must be reached while the balance becomes biased toward cell death through HSV-dependent apoptosis (HDAP). So, it is not surprising that uncontrolled expression of p53 during the virus replication and the increased levels of apoptosis will result in premature host cell death [15–17].

The above explanations remain speculative because, to our knowledge, only one HSV1-p53 based study has been reported [53]. Therefore, the results obtained in our study should be interpreted cautiously.

While the chicken chorioallantoic membrane (CAM) can be a suitable alternative system for isolating ΔUL39/Δγ34.5/HSV-p53, there are several challenges and issues that must be considered for successful use of the CAM model. Firstly, accurately performing this protocol requires training and practice, with egg contamination being the most common problem. Secondly, only low-titer inoculums of the virus can induce discrete pocks on the CAM, while high-titer virus inoculums result in confluent lesions (Fig 4) [38]. Numerous eggs might be required for the isolation of recombinant viruses. In our research, we observed a drop in viral titer in pock lesions 48 hours post-infection, despite the progression of cell proliferation in the virus-infected CAM. This suggests that interferon may be a limiting factor [45]. However, it was shown that the infection potential of Δγ34.5/HSV-1, when used as a control virus for CAM, was very different from that observed in cell culture. As a result, much higher MOIs were required.

In our work, however, the control virus (Δγ34.5/HSV1) showed, at least, the virus with a titer of $1 \times 10^2$ pfu/ml may be required for culturing in the new CAM (data not shown) and the resulting viral titer in each pock was significantly below the required level, and we could not obtain enough recombinant virus to present quantitative data and titrate the rescued virus.

To conclude, the CAM can be a promising but challenging model for mass manufacturing of recombinant viruses such as HSV-1-P53 which are not able to replicate in common cell lines, but whether restoration of wild-type P53 activity by HSV-1 oncolytic would be a potential approach for triggering the p53-mediated pro-apoptotic and enhancing the oncolytic potency in almost all human cancers, deserves further studies.

## Supporting information

**S1 Fig. Sequencing of a pCas-UL39 gRNA expression vector. Chromatograms corresponding to cloned UL39 gRNA.**
(PDF)

**S2 Fig. Schematic diagram of the shuttle vector construction for homologous ecombination.**
(PDF)

**S1 Table. List of primers used in this study, related to the experimental procedures.**
(PDF)

**S2 Table. The sgRNA library targeting the HSV-1 genome.**
(XLSX)

**S1 Raw images.**
(PDF)

## Acknowledgments

We hereby thank the staff of Virology Department and Laboratory of Regenerative Medicine and Biomedical Innovations (Pasteur Institute of Iran).

## Author Contributions

**Conceptualization:** Hosein Shahsavarani, Alijan Tabarraei, Mohammad Ali Shokrgozar, Ladan Teimoori-Toolabi.

**Data curation:** Mishar Kelishadi.

**Formal analysis:** Mishar Kelishadi.

**Investigation:** Mishar Kelishadi.

**Methodology:** Mishar Kelishadi.

**Project administration:** Mishar Kelishadi.

**Supervision:** Kayhan Azadmanesh.

**Validation:** Mishar Kelishadi, Kayhan Azadmanesh.

**Visualization:** Mishar Kelishadi, Kayhan Azadmanesh.

**Writing – original draft:** Mishar Kelishadi.

**Writing – review & editing:** Kayhan Azadmanesh.

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
