## [Decision Letter · Decision Letter 0]

21 Jun 2023

PONE-D-23-14296The chicken chorioallantoic membrane model for isolation of CRISPR/cas9-based HSV-1 mutant expressing tumor suppressor p53PLOS ONE

Dear Dr. Kelishadi,

Thank you for submitting your manuscript to PLOS ONE. After careful consideration, we feel that it has merit but does not fully meet PLOS ONE’s publication criteria as it currently stands. Therefore, we invite you to submit a revised version of the manuscript that addresses the points raised during the review process.

We look forward to receiving your revised manuscript.

Kind regards,

Arunava Roy, Ph.D.

Academic Editor

PLOS ONE

“This study was funded as Ph.D. student project by Pasteur Institute of Iran (Grant Number: TP-9460)”

Reviewers' comments:

Reviewer's Responses to Questions

**Comments to the Author**

1. Is the manuscript technically sound, and do the data support the conclusions?

Reviewer #1: Yes

Reviewer #2: Partly

2. Has the statistical analysis been performed appropriately and rigorously? 

Reviewer #1: No

Reviewer #2: N/A

3. Have the authors made all data underlying the findings in their manuscript fully available?

Reviewer #1: Yes

Reviewer #2: Yes

4. Is the manuscript presented in an intelligible fashion and written in standard English?

Reviewer #1: Yes

Reviewer #2: No

5. Review Comments to the Author

Reviewer #1: A well-written manuscript based on one of the brilliant studies involving good molecular biology practices! However, I’d like to express some general concerns regarding insufficient model system (CAM while good one, seems primitive) broad incompatibility to demonstrate purposes and validity of this study and testing of your hypothesis. The following are some of my comments/suggestions for minor modifications.

Abstract

1. Please incorporate a statement regarding relevance of this piece of research for cancer-specific treatment.

2. In addition, the inability to choose additional models sounds more of a limiting factor rather than strength in terms of disease specificity and cancer heterogeneity. Could you please clarify and incorporate justification here or at the introductory part?

Introduction

1. Please maintain one type of expansion for HSV-1 abbreviation (at its first occurrence).

Discussion

1. Lack of additional phenotypic characterization experiments and usage of cancer specific cell lines and in vivo models may weaken the hypothesis that ΔUL39/Δγ34.5/HSV1-p53 mutant could induce apoptosis in specific cancer cell lines. Please justify.

Tables and Figures

1. Please follow similar patterns to represent primers.

2. Please denote panel names a, b, c, d on images for Figure 3.

3. Expand legend notes for Figure 9. Consider “Representative images..” instead “Representative results..” on the legend title.

4. Figure 10 message seems redundant. Please add what new information Figure 10 could bring to the manuscript.

Reviewer #2: PONE-D-23-14296

"The chicken chorioallantoic membrane model for isolation of CRISPR/cas9-based HSV-1 mutant expressing tumor suppressor p53” by Kelishadi et al”

This manuscript highlights the development of an improved method for rescuing difficult oncolytic viruses using the chorioallantoic membrane (CAM). The introduction provides a concise background on oncolytic viruses as a novel cancer treatment modality. It highlights the use of engineered HSV-1 in phase III clinical trials and emphasizes the need for improved on selectivity and oncotoxicity. The introduction mentions the manipulation of the UL39 gene and the insertion of the EGFP-p53 expression cassette using CRISPR/Cas9-mediated editing to enhance the properties of HSV-1. Furthermore, it introduces the use of the CAM model for isolating the ΔUL39/Δγ34.5/HSV1-p53 mutant and its comparison with the parent Δγ34.5/HSV-1 in vitro.

The methods section lacks specific details regarding the experimental procedures used. It should provide a step-by-step explanation of the CRISPR/Cas9-mediated editing of the UL39 gene and the insertion of the EGFP-p53 expression cassette. Additionally, the process of isolating the ΔUL39/Δγ34.5/HSV1-p53 mutant using the CAM model should be described in more detail, including the specific techniques and controls employed.

The results section should provide a more detailed explanation of the phenotypic characterization of ΔUL39/Δγ34.5/HSV1-p53-infected cells compared to the parent Δγ34.5/HSV-1 in vitro. It should include quantitative data, such as foci count and viral titer, to support the observations. Additionally, the results should be presented in a clear and organized manner, and the figures should be properly labeled and explained.

The conclusion provided in the abstract is consistent with the findings of the study, highlighting the potential of the CAM model for isolating recombinant viruses like HSV-1-P53 that cannot replicate in cell lines due to exogenous p53-induced cell death. However, it would be beneficial to summarize the main findings and their implications in a more comprehensive manner, drawing connections to the potential clinical applications of the engineered HSV-1.

Overall, the manuscript shows promise but requires further revisions and improvements. The methods section should provide more specific details, and the results section should present quantitative data and be more organized. By addressing these issues, the study's contribution to the field of oncolytic viruses and cancer treatment can be better appreciated.

Specific comments:

Abstract:

The abstract provides a concise overview of the study, highlighting the phenotypic characterization of ΔUL39/Δγ34.5/HSV1-p53-infected cells in comparison to the parent Δγ34.5/HSV-1 in vitro. However, there is a lack of available data in the main results section to support the claims made in the abstract.

Introduction:

The introduction sets the stage for the study, but there are several areas that require improvement.

Firstly, the abbreviation "ICP" should be expanded to "immediate early protein" and followed by a brief explanation of its concept for better reader understanding. Additionally, the citation mentioned in line 62 (Reference 17) should be explained further, focusing on the role of ICP0 in the context of HSV oncolytic virus.

Materials and Methods:

The materials and methods section needs attention regarding several aspects.

Line 127-129 contains an incomplete sentence that needs to be revised for clarity.

Moreover, line 133 includes irrelevant references for the state CAM utility and should be removed.

Throughout the manuscript, the gene name "UL39" (for example, in line 145) should be consistently checked for accuracy.

Line 147 requires the addition of a reference for CRISPR technology.

In line 219, it should be mentioned which IRES is used in the construct.

Furthermore, a control virus expressing only GFP should be included to test the CAM rescue and testing the role of p53 in the oncolytic event mentioned in the paper.

Line 223 needs clarification on whether it refers to the co-seeding of Vero and BHK-21 cells. Lines 244 and 255 should specify the medium used, as Figure 2 (9.2) states that selected pocks are taken into the culture medium. Additionally, it is unclear whether vertexing or trypsinization is the actual step taken for the mentioned procedure.

Results:

The results section is poorly written, lacking detailed explanations of the observations. Furthermore, figure labels are not properly labeled, requiring improvement. One major drawback of the paper is the predominance of qualitative results rather than quantitative data. Foci count or peak viral titer obtained for viral rescue from the cell line needs to be compared with the CAM route.

Line 296: Figure 3 should be marked with panels a, b, c, and d.

Line 304: Microscopic details should be added to the Materials and Methods, including information about the microscope used, the objective, magnification, and other relevant details.

Line 308: Specify the type of filter used.

Line 314: Specify the microscopy settings differences between panels a and b.

Line 326: Figure 7 has low resolution and is difficult to read. A higher-resolution image should be provided.

Line 327: Change "bold" to "normal" font.

Line 335: Clarify where the data comparing infected ΔUL39/Δγ34.5/HSV-p53 cells with the parental Δ34.5/HSV-1 is displayed. Figures 8 and 9 show data for ΔUL39/Δγ34.5/HSV-p53, but the data for Δ34.5/HSV-1 is not clearly presented.

Line 336: Replace "stronger cell-killing" with a more scientifically appropriate term, such as "cytolytic" ability.

Lines 339-341: Elaborate on the observations of viral infection in cancerous and non-cancerous cell lines.

Line 344: Explain why the 72-hour time point shows a more intense signal compared to the 5-day time point.

Line 348: Clarify the labeling of the "Bright field + fluorescence overlay" in the MDA-MB-468 panel, and specify the magnification of the microscope used for this imaging. Additionally, the overlay should show yellow (red + green), but the last panel only shows a green overlay in bright field.

Discussion:

Line 380: Separate references 33 and 35.

Line 386: Provide evidence or data to support the statement made.

Lines 430-431: Therefore, the results obtained in our study should be interpreted cautiously.

Line 442: Provide data to support the claim that Δγ34.5/HSV-1, as a control virus for CAM, exhibits different behavior compared to cell culture, and that the MOIs should be much higher.

In conclusion, this manuscript requires significant revisions and improvements to address the issues outlined above. The authors should focus on providing more detailed and quantitative results, properly labeling figures, and ensuring consistency throughout the text. Additionally, clarifications and additional information are needed in various sections to enhance the scientific rigor of the study.

6. PLOS authors have the option to publish the peer review history of their article (what does this mean?). If published, this will include your full peer review and any attached files.

Reviewer #1: **Yes: **Titto Augustine

Reviewer #2: No

---

## [Author Response · Author response to Decision Letter 0]

16 Aug 2023

Response letter

Date: August 4, 2023

To: Editor-in-Chief of PLOS ONE

Subject: submission of the revised manuscript PONE-D-23-14296, titled as: “The chicken chorioallantoic membrane model for isolation of CRISPR/cas9-based HSV-1 mutant expressing tumor suppressor p53"

Dear Dr. Arunava Roy,

We would like to thank the reviewers for their very helpful comments and suggestions that have allowed us to improve our study (Manuscript PONE-D-23-14296). We have been able to incorporate changes to reflect most of the suggestions provided by the reviewers. We have highlighted the changes within the manuscript. Correction of grammatical errors and English improvement were carried done by a native English-speaking editor as suggested (Red words). So we hope it now matches the journal standard.

Here is a point-by-point response to the reviewers' comments and concerns. We hope this will be satisfactory and meet with your expectations for publication in PLOS ONE.

The academic editor and reviewers comments and responses

The academic editor

Query Author’s response

 Thank you for this option. It sounds interesting. Is it possible to submit the protocol with a slightly different title? This manuscript is for obtaining the Ph.D degree of the first author, and 2 DOIs with the same title will not serve this goal for her. 

Journal requirements 

Query Author’s response

1- Please ensure that your manuscript meets PLOS ONE's style requirements, including those for file naming. The PLOS ONE style templates can be found at https://journals.plos.org/plosone/s/file?id=wjVg/PLOSOne_formatting_sample_main_body.pdf and https://journals.plos.org/plosone/s/file?id=ba62/PLOSOne_formatting_sample_title_authors_affiliations.pdf

 The comments in the manuscript were addressed according to PLOS ONE's style.

2- Thank you for stating the following financial disclosure:

“This study was funded as Ph.D. student project by Pasteur Institute of Iran (Grant Number: TP-9460)”

Please include your amended statements within your cover letter; we will change the online submission form on your behalf. The amended statements were included within our cover letter and the text of the manuscript:

a) This study was funded as Ph.D. student project by Pasteur Institute of Iran (Grant Number: TP-9460)

b) The funder had no role in study design, data collection and analysis, decision to publish, or preparation of the manuscript.

C&d) The authors received no specific funding for this work.

3- We note that you have stated that you will provide repository information for your data at acceptance. Should your manuscript be accepted for publication, we will hold it until you provide the relevant accession numbers or DOIs necessary to access your data. If you wish to make changes to your Data Availability statement, please describe these changes in your cover letter and we will update your Data Availability statement to reflect the information you provide. The amended statement was included in our cover letter:

All data underlying the findings are in the manuscript fully available and reported in the text.

4. PLOS requires an ORCID iD for the corresponding author in Editorial Manager on papers submitted after December 6th, 2016. Please ensure that you have an ORCID iD and that it is validated in Editorial Manager. To do this, go to ‘Update my Information’ (in the upper left-hand corner of the main menu), and click on the Fetch/Validate link next to the ORCID field. This will take you to the ORCID site and allow you to create a new iD or authenticate a pre-existing iD in Editorial Manager. Please see the following video for instructions on linking an ORCID iD to your Editorial Manager account: https://www.youtube.com/watch?v=_xcclfuvtxQ A mistake has happened during the submission of this manuscript. It is submitted by M. Kelishadi instead of K. Azadmanesh (which is stated as the corresponding author in the manuscript itself). Would you please guide us how we can correct this? The ORCID ID for the corresponding author is:

Kayhan Azadmanesh 

https://orcid.org/0000-0002-7165-9043

REVIEWER 1

Query Author’s response

Abstract:

1. Please incorporate a statement regarding relevance of this piece of research for cancer-specific treatment. 

 The following sentence was added to the text (Lines 21 and 22 of the revised manuscript): 

Previous studies showed that design of OV therapy combined with p53 gene therapy increases the anti-cancer activities of OVs. 

Abstract:

2. In addition, the inability to choose additional models sounds more of a limiting factor rather than strength in terms of disease specificity and cancer heterogeneity. Could you please clarify and incorporate justification here or at the introductory part? Because our early attempts failed for isolation of the recombinant ΔUL39/Δ γ34.5/ HSV-p53 by cultivation of the transfected/infected viral supernatant in cell lines, we performed a proof of concept study to investigate whether the CAM model can be promising for isolating of the HSV-1 mutants.

So we added “a proof of concept study” in line 85 of the revised manuscript .

 Introduction:

1. Please maintain one type of expansion for HSV-1 abbreviation (at its first occurrence).

 We have addressed the comment in the text (Lines 20, 39 and 47 of the revised manuscript).

Discussion:

1. Lack of additional phenotypic characterization experiments and usage of cancer specific cell lines and in vivo models may weaken the hypothesis that ΔUL39/Δγ34.5/HSV1-p53 mutant could induce apoptosis in specific cancer cell lines. Please justify.

 The manuscript highlights the development of an improved method for isolation of difficult oncolytic viruses using the chorioallantoic membrane (CAM). The main question of this study was how to produce such a recombinant virus and not its application in cancer cells. 

Although cultivation of ΔUL39/Δ γ34.5/HSV-p53 mutant in these cell lines was associated with rounding of cells in early infection without viral cell-to-cell spread and loss of adherence to the monolayer, the number of live infected cells after 48 hours was too low to enable us study the specific mechanism of cell death. 

To clarify the aim of this study and its justification, the last paragraph of the discussion was changed as follows (Lines 346-349 of the revised manuscript: 

To conclude, the CAM can be a promising but challenging model for mass manufacturing of recombinant viruses such as HSV-1-P53 which are not able to replicate in common cell lines, but whether restoration of wild-type P53 activity by HSV-1 oncolytic would be a potential approach for triggering the p53-mediated pro-apoptotic and enhancing the oncolytic potency in almost all human cancers, deserves further studies.

This manuscript has no claim regarding how well this virus may work in vivo yet.

Tables and Figures:

1. Please follow similar patterns to represent primers.

 The table 1 was corrected accordingly (The Line 123 of the revised manuscript).

Tables and Figures:

2. Please denote panel names a, b, c, d on images for Figure 3.

 Figure 3 was corrected accordingly.

Tables and Figures:

3. Expand legend notes for Figure 9. Consider “Representative images.” instead“Representative results.” on the legend title. The legend of Figure 9 was corrected accordingly (The Line 277-280 of the revised manuscript).

Tables and Figures:

4. Figure 10 message seems redundant. Please add what new information Figure 10 could bring to the manuscript.

 According to reviewer comment, the graph was removed (The Line 314 of the revised manuscript).

REVIEWER 2

Query Author’s response

Abstract: 

The abstract provides a concise overview of the study, highlighting the phenotypic characterization of ΔUL39/Δγ34.5/HSV1-p53-infected cells in comparison to the parent Δγ34.5/HSV-1 in vitro. However, there is a lack of available data in the main results section to support the claims made in the abstract.

 Thank you for your detailed reveiw and comments, this claim is based on the evidence provided in figures 3 and 9 . Since there were few infected living cells after 48 hours in each plate, we were not able to study the mechanism of cell death in more detail.

Introduction: 

1-The introduction sets the stage for the study, but there are several areas that require improvement. Firstly, the abbreviation "ICP" should be expanded to "immediate early protein" and followed by a brief explanation of its concept for better reader understanding. The first time ICP is used (Line 41), its complete name is added to the text. If further explanation is needed in this context, would you please let us know?

Introduction: 

2-Additionally, the citation mentioned in line 62 (Reference 17) should be explained further, focusing on the role of ICP0 in the context of HSV oncolytic virus. Thank you for your comment, reference 17 was not relevant to this paragraph, so it was deleted and added to next pharagraghs that explain about the role of ICP0 in the context of HSV oncolytic virus (Lines 50-54 of the revised manuscript).

Materials and Methods:

3-The materials and methods section needs attention regarding several aspects.

Line 127-129 contains an incomplete sentence that needs to be revised for clarity.

 The sentence was re-written for clarification (Lines 98-99 of the revised manuscript).

4-Moreover, line 133 includes irrelevant references for the state CAM utility and should be removed. These references are to confirm this sentence:”special permission for egg experiments was not required” (Lines 104-105) it means there is no need to an ethical permission for this experiment. 

Throughout the manuscript, the gene name "UL39" (for example, in line 145) should be consistently checked for accuracy.

 The comment was addressed in the text (Lines 113, 126, 127, 141, 143, 144, 154, 219 and 255- 258 of the revised manuscript).

Line 147 requires the addition of a reference for CRISPR technology. The website of the protocol was added to the text (Lines 115-117 of the revised manuscript).

In line 219, it should be mentioned which IRES is used in the construct. 

 The comment was addressed in the text (Lines 129, 141 and 168 of the revised manuscript).

Furthermore, a control virus expressing only GFP should be included to test the CAM rescue and testing the role of p53 in the oncolytic event mentioned in the paper.

 Thank you for your comment, As described, this study was a Ph.D. project and the time and budget constraints did not allow us to do more. But following your comment,we constructed the ΔUL39 (green)/Δ 34.5 (red) HSV1-EGFP and the ΔUL39 (green)/Δ 34.5 (red) HSV1-EGFP-p53 using the conventional homologous recombination method. HSV1-GFP replicated in BHK cell line but HSV1-P53 replication in BHK cell line was limited to a single cell (Figure 8 of the revised manuscript). Due to time constraints we are not able to repeat all other cell lines tested for ΔUL39 (green)/Δ 34.5 (red) HSV1-EGFP-p53. We hope at this stage this would be satisfactory for the reviewer.

Line 223 needs clarification on whether it refers to the co-seeding of Vero and BHK-21 cells. The procedure worked for both vero and BHK-21 cells, but since most of the experiments were done on vero cells, BHK-1 was deleted from the text. (Line 172 of the revised manuscript).

Lines 244 and 255 should specify the medium used, as Figure 2 (9.2) states that selected pocks are taken into the culture medium. DMEM was used for this procedure. It is now clarified in the text (Lines 186 and 197 of the revised manuscript, Legend 2).

Additionally, it is unclear whether vertexing or trypsinization is the actual step taken for the mentioned procedure.

 Both vortexing and trypsinization are the critical steps in the CAM experiment. The pocks dissection, enzymatic digestion, and mechanical dissociation are three significant steps leading to the degradation of the extracellular matrix and the isolation of the viruses, allowing the generation of high viral titer. 

The text was clarified according to this explanation (line 189 of the revised manuscript).

Results:

The results section is poorly written, lacking detailed explanations of the observations. Furthermore, figure labels are not properly labeled, requiring improvement. One major drawback of the paper is the predominance of qualitative results rather than quantitative data. Foci count or peak viral titer obtained for viral rescue from the cell line needs to be compared with the CAM route.

 In our work, the resulting viral titer in each pock were significantly low. As well, a drop in viral titer were seen in pock lesions 48 h post-infection, despite cell proliferation progression in the virus-infected CAM, Given that in all of ΔUL39/Δγ34.5/HSV1-p53-infected cell lines, the recombinant virus replication was limited to a single cell, many pocks were required .However, The control virus (Δγ34.5/HSV1) showed , at least, the virus with a titer of 1 × 102 pfu/ml may be required for culturing in the new CAM and we could not obtain enough recombinant virus to present quantitative data and titrate the rescued virus (line 343-345 of the revised manuscript).

Line 296: Figure 3 should be marked with panels a, b, c, and d. The comment was addressed in the figure 3 (line 230 of the revised manuscript).

Line 304: Microscopic details should be added to the Materials and Methods, including information about the microscope used, the objective, magnification, and other relevant details.

 The brand and model of the microscope was added to the text (Line 174 of the revised manuscript). Furthermore the scale bar of each photo was added through the document in relevant sites.

Line 308: Specify the type of filter used.

 The comment was addressed in the text (Line 236-237 of the revised manuscript, figure 4).

Line 314: Specify the microscopy settings differences between panels a and b. This sentence was modified as follow:

Macroscopic picture of high titer virus -infected CAM (a). Microscopic picture of high titer virus -infected CAM (Magnification × 100) (b). (Line 243-244 of the revised manuscript, figure 5)

Line 326: Figure 7 has low resolution and is difficult to read. A higher-resolution image should be provided. Figure 7 was modified accordingly. 

Line 327: Change "bold" to "normal" font. The comment was addressed in the text (Legend 7, line 256 of the revised manuscript).

Line 335: Clarify where the data comparing infected ΔUL39/Δγ34.5/HSV-p53 cells with the parental Δ34.5/HSV-1 is displayed. Figures 8 and 9 show data for ΔUL39/Δγ34.5/HSV-p53, but the data for Δ34.5/HSV-1 is not clearly presented. Thank you for your comment, As described, this study was a Ph.D. project and the time and budget constraints did not allow us to do more. But following your comment,we constructed the ΔUL39 (green)/Δ 34.5 (red) HSV1-EGFP and the ΔUL39 (green)/Δ 34.5 (red) HSV1-EGFP-p53 using the conventional homologous recombination method. HSV1-GFP replicated in BHK cell line but HSV1-P53 replication in BHK cell lin Thank you for your comment, As described, this study was a Ph.D. project and the time and budget constraints did not allow us to do more. But following your comment,we constructed the ΔUL39 (green)/Δ 34.5 (red) HSV1-EGFP and the ΔUL39 (green)/Δ 34.5 (red) HSV1-EGFP-p53 using the conventional homologous recombination method. HSV1-GFP replicated in BHK cell line but HSV1-P53 replication in BHK cell line was limited to a single cell (Figure 8 of the revised manuscript). Due to time constraints we are not able to repeat all other cell lines tested for ΔUL39 (green)/Δ 34.5 (red) HSV1-EGFP-p53. We hope at this stage this would be satisfactory for the reviewer.e was limited to a single cell (Figure 8 of the revised manuscript). Due to time constraints we are not able to repeat all other cell lines tested for ΔUL39 (green)/Δ 34.5 (red) HSV1-EGFP-p53. We hope at this stage this would be satisfactory for the reviewer.

Line 336: Replace "stronger cell-killing" with a more scientifically appropriate term, such as "cytolytic" ability. The comment was addressed in the text (Line 266 of the revised manuscript).

Lines 339-341: Elaborate on the observations of viral infection in cancerous and non-cancerous cell lines. Since we used cell lines, all of them are immortal and have some cancerous features. However they have different origins ( human- non-human) and wildtype- mutant P53 proteins. Based on these distinctions the text was re-written (Lines 262-265 of the revised manuscript).

Line 344: Explain why the 72-hour time point shows a more intense signal compared to the 5-day time point.

 We believe the difference in the brightness of those 2 pictures was mostly due to different exposure and acquisition settings of the camera, since the back ground brightness of the control graph was also higher. However, based on an earlier comment by you we preferred to change the control picture with the picture of ΔUL39 (green)/Δ 34.5 (red) HSV1-EGFP. The brightness of both coulours in cells infected with this virus are much more similar to the test virus.

Line 348: Clarify the labeling of the "Bright field + fluorescence overlay" in the MDA-MB-468 panel, and specify the magnification of the microscope used for this imaging. Additionally, the overlay should show yellow (red + green), but the last panel only shows a green overlay in bright field. The comment was addressed in figure 9 and legend 9 (Lines 277-280 of the revised manuscript).

Discussion:

Line 380: Separate references 33 and 35.

 The comment was addressed in the text (Line 305 of the revised manuscript).

Line 386: Provide evidence or data to support the statement made:

Therefore, the results obtained in our study should be interpreted cautiously. Since there is only one other report of HSV1-p53 in the literature, we believe our speculations have not enough supportive evidence and needs further evaluation by other groups. However, this does not affect the method we developed for hard to produce recombinant HSV-1.

Line 442: Provide data to support the claim that Δγ34.5/HSV-1, as a control virus for CAM, exhibits different behavior compared to cell culture, and that the MOIs should be much higher.

 The comment was addressed in the text (Line 343 of the revised manuscript).

In conclusion, this manuscript requires significant revisions and improvements to address the issues outlined above. The authors should focus on providing more detailed and quantitative results, properly labeling figures, and ensuring consistency throughout the text. Additionally, clarifications and additional information are needed in various sections to enhance the scientific rigor of the study.

 We thank the reviewers for their suggestions for improving the presentation of our manuscript.

---

## [Decision Letter · Decision Letter 1]

11 Sep 2023

The chicken chorioallantoic membrane model for isolation of CRISPR/cas9-based HSV-1 mutant expressing tumor suppressor p53

PONE-D-23-14296R1

Dear Dr. Azadmanesh,

We’re pleased to inform you that your manuscript has been judged scientifically suitable for publication and will be formally accepted for publication once it meets all outstanding technical requirements.

Kind regards,

Arunava Roy, Ph.D.

Academic Editor

PLOS ONE

Additional Editor Comments (optional):

Reviewers' comments:

Reviewer's Responses to Questions

**Comments to the Author**

1. If the authors have adequately addressed your comments raised in a previous round of review and you feel that this manuscript is now acceptable for publication, you may indicate that here to bypass the “Comments to the Author” section, enter your conflict of interest statement in the “Confidential to Editor” section, and submit your "Accept" recommendation.

Reviewer #2: All comments have been addressed

2. Is the manuscript technically sound, and do the data support the conclusions?

Reviewer #2: Yes

3. Has the statistical analysis been performed appropriately and rigorously? 

Reviewer #2: N/A

4. Have the authors made all data underlying the findings in their manuscript fully available?

Reviewer #2: Yes

5. Is the manuscript presented in an intelligible fashion and written in standard English?

Reviewer #2: Yes

6. Review Comments to the Author

Reviewer #2: The chicken chorioallantoic membrane model for isolation of CRISPR/cas9-based HSV-1 mutant expressing tumor suppressor p53

The authors have made the suggested revisions and the comments were addressed in the response to comments. The revised manuscript shows improvements in the contents, details and organization of the manuscript.

Please see the minor specific comments on the revised manuscript

Specific comments

Line 51: Please clarify what is the conflicting data on apoptosis with required reference.

Line 72: VSV: Expand “Vesicular Stomatitis Virus”

Line 126, 192, 218,233, 278,: replace the word “construction” with” rescue” or “generation”. Construction is usually used for describing vectors or plasmids.

7. PLOS authors have the option to publish the peer review history of their article (what does this mean?). If published, this will include your full peer review and any attached files.

Reviewer #2: No

---

## [Editor Report · Acceptance letter]

13 Oct 2023

PONE-D-23-14296R1 

The chicken chorioallantoic membrane model for isolation of CRISPR/cas9-based HSV-1 mutant expressing tumor suppressor p53 

Dear Dr. Azadmanesh:

I'm pleased to inform you that your manuscript has been deemed suitable for publication in PLOS ONE. Congratulations! Your manuscript is now with our production department. 

Kind regards, 

on behalf of

Dr. Arunava Roy 

Academic Editor

PLOS ONE